# Magnetoelectric Properties of Ni-PZT-Ni Heterostructures Obtained by Electrochemical Deposition of Nickel in an External Magnetic Field

Natalia Poddubnaya [1] , Dmitry Filippov [2,*] , Vladimir Laletin [1], Aliaksei Aplevich [3] and Kazimir Yanushkevich [3]

[1] Institute of Technical Acoustics, National Academy of Sciences of Belarus, 210009 Vitebsk, Belarus; poddubnaya.n@rambler.ru (N.P.)
[2] Polytechnic Institute, Yaroslav the Wise Novgorod State University, 173003 Veliky Novgorod, Russia
[3] Scientific and Practical Materials Research Center, Institute of Semiconductor and Solid State Physics, National Academy of Sciences of Belarus, 220072 Minsk, Belarus
* Correspondence: dmitry.filippov@novsu.ru

**Abstract:** This paper studied the influence of external electric and magnetic fields on the magneto-electric properties of layered structures of metal-piezoelectric-metal. The structures under study had the shape of a square 4 mm wide and were obtained in two steps: first, by the chemical deposition of nickel with a thickness of 0.5 μm, and then by the electrochemical deposition of nickel with a thickness of 50 μm on each side onto a lead zirconate–lead titanate substrate. Electrochemical deposition was carried out without a magnetic field on both non-polarized and polarized ceramics. Electrochemical deposition was also carried out in a magnetic field on a non-polarized and polarized PZT ceramic substrate. A magnetic field of 500 Oe at electrochemical deposition was applied in all cases in the direction of structure polarization. The maximum ME voltage coefficient 300 mV/(cmOe) was obtained at transverse orientation at bias magnetic field near 20 Oe.

**Keywords:** magnetoelectric effect; heterostructure; electrochemical deposition; magnetoelectric voltage coefficient

## 1. Introduction

The relationship between magnetic, structural, and electrical parameters of composite structures with magnetoelectric properties is being actively studied by an increasing number of scientific research centers. This is justified by many promising devices based on the conversion of magnetic field energy into electrical energy through the transfer of elastic deformation, devices for recording and storing information, etc. [1–6]. There are many works devoted to obtaining giant values of the ME coefficient in bulk and layered structures. The most studied effects are in composites based on PZT and materials with high magnetostriction—Metglas, Permendur, Terfenol-D, etc. [4,7]. Many works are devoted to resonance effects, which make it possible to achieve an ME coefficient of up to several V/cmOe [6,8–12]. Research has been reported in complex nanostructures, where the magnetic phase is distributed in a dedicated volume (particles, fibers, and columns) of a matrix with piezoelectric properties [13–16]. Several experimental and theoretical studies confirm the advantage of layered composites over bulk ones [17]. Their main advantage lies in the possibility of using materials with maximum magneto- and piezostrictive characteristics. At the same time, alloys based on rare earth metals exhibit effects in high magnetic fields, which is difficult to apply in practice. The use of heterostructures based on soft magnetic materials opens their wide use for creating double sensors and memory elements [18,19]. Studies have been carried out indicating the possibility of increasing the magnetostrictive properties of soft magnetic materials using an external magnetic field in the process of structure formation [11–23]. In this work, the influence of external magnetic and electric fields on the magnetoelectric properties of layered composite structures was considered.

## 2. Materials and Methods

Samples of powdered ceramics PZT composition $PbZr_{0.45}Ti_{0.55}O_3$ were produced by traditional high-temperature sintering technology at 1280 °C. They were made in the form of squares with a side length of 4 mm and a height of 400 μm. Nickel electrodes were applied by chemical metallization. The entire batch of samples was divided into four identical parts. The first part was polarized, then passed to electrochemical deposition (el/P). The second part was polarized, then subjected to electrochemical deposition in a magnetic field (P/el+H). The third and fourth parts were first precipitated in a magnetic field, then without it, and then polarized (el/P and el+H/P, respectively).

In all cases, polarization was carried out in silicone oil under the action of an electric field of 4 kV/mm at a temperature of 150 °C for an hour. Chemical deposition was carried out from a solution of nickel sulfate ($NiSO_4$, $NH_4Cl$, $NaC_6H_5O_7$, $NaH_2PO_2$ and $NH_4OH$) at pH~8–9, t~70 °C onto the ceramic surface previously degreased in a 10% KOH solution after sensitization with stannous chloride, activation with palladium chloride solution, and acceleration with nickel hypophosphite. Electrochemical deposition of nickel was carried out from the solution of nickel sulfamate, six aqueous nickel chloride, and boric acid with the addition of saccharin at pH~3.5, t~40 °C and a current of 6–8 A/dm$^2$ with continuous stirring. The nickel coating thickness on each side was ~50 μm. In all cases, the magnetic field was directed along the polarization and was perpendicular to the sample plane (Figure 1).

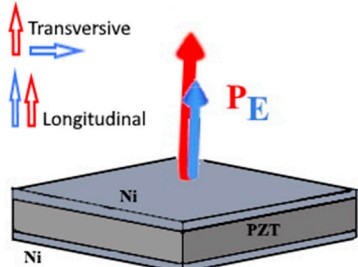

**Figure 1.** Schematic diagram of the heterostructure.

The dependence of the magnitude of the ME coefficient $\alpha_E(H)$ of layered structures under the action of an external constant magnetic field, which varies from $-200$ kA/m to 200 kA/m, has been experimentally studied. The value of the ME coefficient was determined from the voltage ($U$) generated by the sample, taking into account the thickness of the composite ($h$) and the magnitude of the alternating magnetic field ($H$) according to the formula: $\alpha_E = \frac{U}{h \cdot H}$.

An alternating magnetic field with a frequency of 1 kHz and a strength of about 80 kA/m was used. In the case when the applied magnetic field was oriented perpendicular to the polarization of the structure, we are referring to the study of the transverse ME effect. When there was an external field co-directional with the polarization orientation, we are referring to the longitudinal ME effect. The effect was linearized under the action of a constant magnetic field.

The measurements of the magnetic parameters of the structures were carried out by the ponderomotive method on the unique research equipment of the Scientific-Practical Materials Research Centre of NAS of Belarus in accordance with the method MN 3128-2009. The method is based on the ponderomotive interaction of the recorded magnetic field of a film suspended in a quartz cell in each position and the constant magnetic field of an electromagnet [21]. The measurement error of specific magnetization was 0.005 Am$^2$ kg$^{-1}$; the specific magnetic susceptibility was $10^{-11}$ m$^3$ kg$^{-1}$.

## 3. Experimental Results and Discussion

Comparative data of the transverse ME effect with longitudinal and transverse polarization of layered structures with mass production are shown in Figure 2.

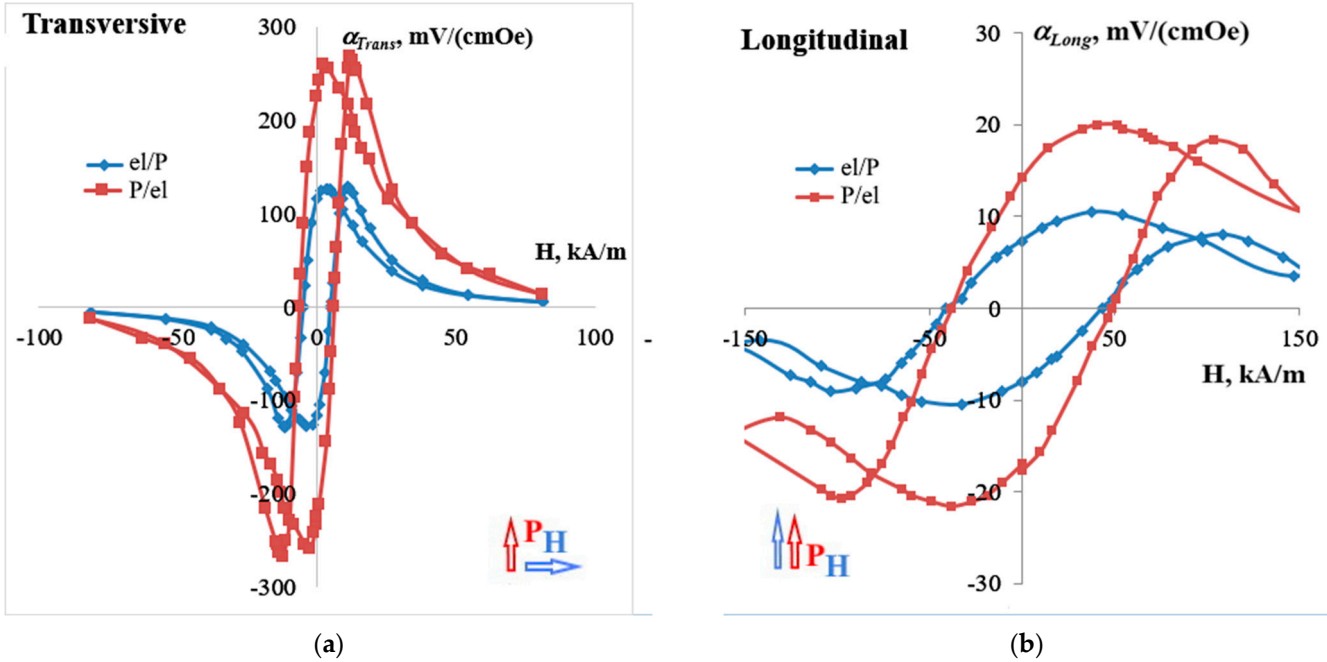

**Figure 2.** Linear magnetoelectric effect in samples coated with nickel with a thickness of 50 μm on each side: (**a**) for transverse orientation; (**b**) for longitudinal orientation.

Experimental data on the ME coefficient of nickel-PZT-nickel layered structures obtained in a magnetic field are shown in Figure 3.

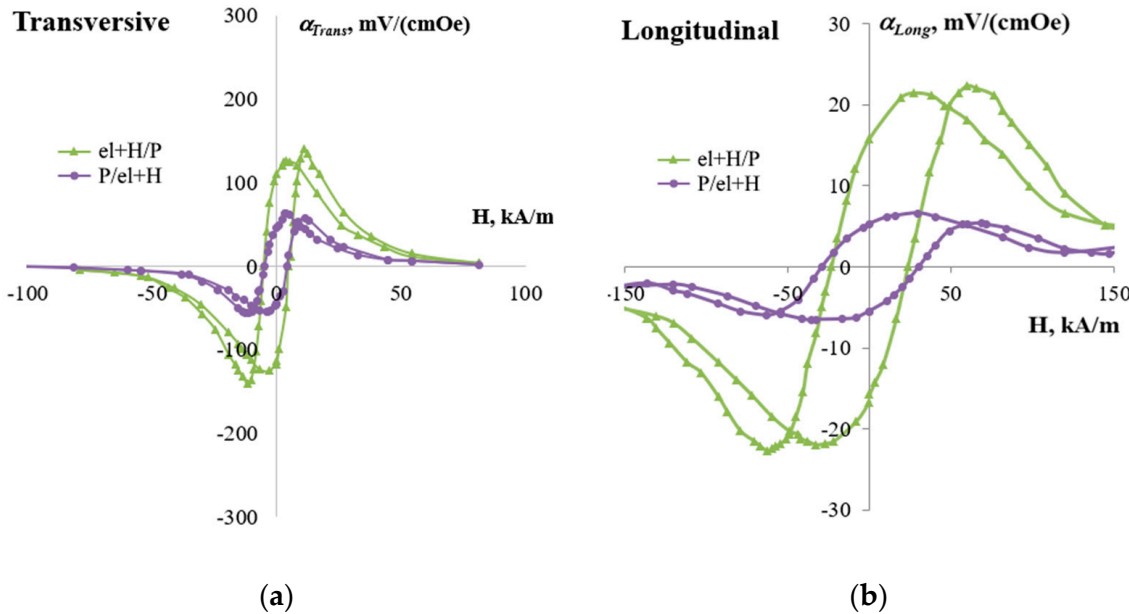

**Figure 3.** Linear magnetoelectric effect in samples coated with nickel with a thickness of 50 μm on each side, obtained in a magnetic field: (**a**) for transverse orientation; (**b**) for longitudinal orientation.

For a film obtained by deposition without a magnetic field on pre-polarized ceramics, a study of the temperature dependence of specific magnetization, magnetic susceptibility, and crystal structure was carried out (Figure 4).

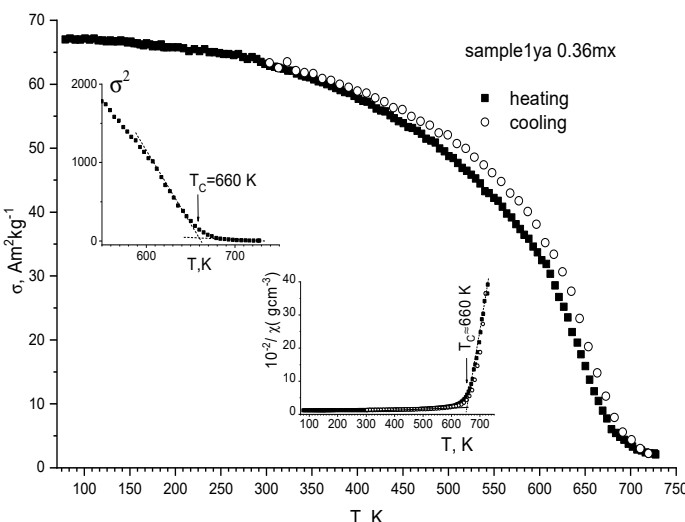

**Figure 4.** Temperature dependencies of the specific magnetization σ = *f*(*T*), obtained by the ponderomotive method in the "heating-cooling" mode in the temperature range 80 K < *T* < 750 K in a magnetic field with induction B = 0.86 Tesla.

For the purpose of an approximate estimate of the value of the Curie temperature $T_C$ of the magnetic phase transition "magnetic order—magnetic disorder", the transition from the ferromagnetic ordering of magnetic moments at temperatures $T < T_C$ to the paramagnetic state at $T > T_C$, the bottom inset of Figure 4 in the same temperature range shows the temperature, the dependence of the reciprocal values of the specific magnetic susceptibility $10^{-2}/\chi$ = f(T), in a magnetic field with the same induction *B*.

Figure 5 shows the data of X-ray diffraction analysis of a nickel coating obtained on polarized ceramics at room temperature in *Cu* $K_a$ radiation.

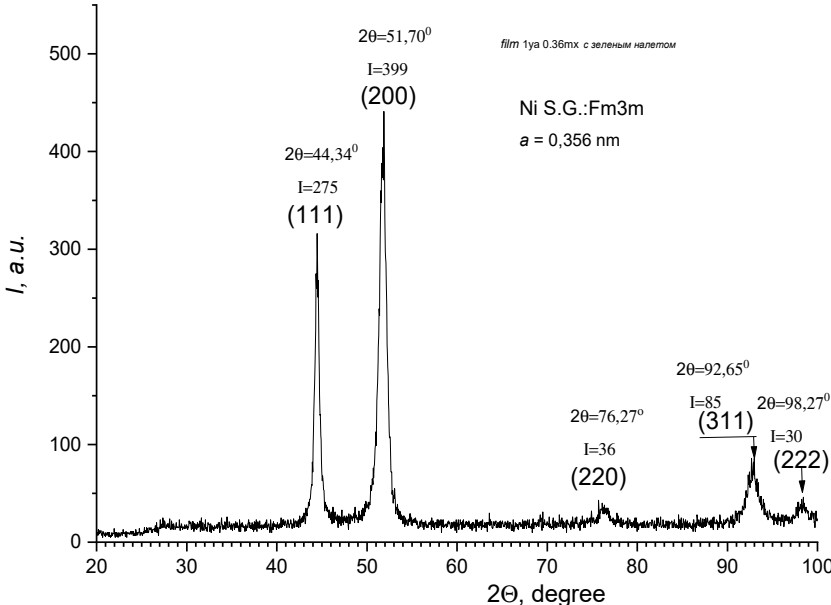

**Figure 5.** X-ray of el/p film. All reflections on the radiograph are indicated in accordance with cards 04-08950, 88-2326 PCPDFWIN v.2 1998 JCPDS.

According to the database card 04-08950, the angular values of these reflections and their intensities are given in Table 1.

**Table 1.** Angular values of reflections and their intensities.

| hkl | 2θ | Intensity |
|-----|-----|-----------|
| 111 | 44.505 | 100 |
| 200 | 51.844 | 43 |
| 220 | 76.366 | 21 |
| 311 | 92.939 | 20 |
| 222 | 98.440 | 7 |

For nickel films separated from the substrates of the P/el, el/P, P/el+H, and el+H/P structures, the specific magnetic susceptibility and specific magnetization were measured at room temperature and different orientations to an external magnetic field with an induction of 0.86 Tesla. The results of the study averaged over three films are presented in Table 2.

**Table 2.** Specific magnetic susceptibility and specific magnetization of nickel films depending on the conditions for obtaining samples and the orientation of the magnetic field during measurements.

| Samples | Magnetic Parameters | P↑↑H↑↑h | Change Characteristic | P↑↑H↑→h |
|---------|--------------------|---------|----------------------|---------|
| El/P | μ (*magnetic permeability*) | 6.28 | | 6.55 |
| | χ (*magnetic susceptibility*) | 13.58 | | 13.05 |
| P/El | μ (*magnetic permeability*) | 6.94 | | 7.13 |
| | χ (*magnetic susceptibility*) | 12.30 | | 11.99 |
| El+H/P | μ (*magnetic permeability*) | 3.74 | | 3.90 |
| | χ (*magnetic susceptibility*) | 22.86 | | 21.89 |
| P/El+H | μ (*magnetic permeability*) | 3.15 | | 3.29 |
| | χ (*magnetic susceptibility*) | 27.10 | | 25.94 |

## 4. Discussion

From Figure 2, it is obvious that the action of the electric field of ceramics polarized before deposition leads to an increase in the ME coefficient, both for longitudinal and transverse polarizations, by a factor of two. The study of the values of the ME coefficient during the deposition of cobalt coatings showed a similar result. On the contrary, the effect of a magnetic field during the production of nickel coatings leads to a significant decrease in the transverse ME coefficient (Figure 4). In this case, with regards to the longitudinal effect, the value of the ME coefficient is higher for pre-polarized structures obtained without

the action of a magnetic field, and lower for polarized blanks metallized in a magnetic field. Therefore, the structure of the P/el film aroused the greatest interest, for which an X-ray diffraction study and the most detailed analysis of the magnetic properties were carried out. According to literature sources [21], the specific magnetization of nickel films at room temperature is $\sigma_s$(287 K) = 55.37 A·m$^2$·kg$^{-1}$, and the Curie temperature $T_C$ = 627.4 K. characteristics of the film substance are much higher: $\sigma_s$(287 K) = 64.25 A·m$^2$·kg$^{-1}$, and the Curie temperature $T_C$ = 660 K. The upper inset of Figure 4, the dependence $\sigma^2 = f(T)$, as well as the first derivative of the values of specific magnetization, make it possible to most accurately determine the value of the Curie temperature. The results of the X-ray experiment of the P/el film showed a significant deformation of the unit cell of the film, which indicates the presence of ferromagnetic ordering, since the specific magnetization and the Curie temperature also increased. Comparing the data in Table 1 with Figure 5, one can see a redistribution of the intensity of reflections, especially (111) and (311). This result of the experiment indicates a significant deformation of the unit cell in the direction of 111 and 100 cubic system. The angular positions of all reflections of the film are shifted to the region of smaller angles: in the direction of 111 by 0.37%, in the direction of 200 by 0.28%, in the direction of 220 by 0.12%, in the direction of 311 by 0.31%, and in the direction of 222 by 0.17%. This also indicates a significant deformation of the crystal cell in different directions. In this case, the distortions of the crystal structure along different axes reach a difference of 2–3 times. Such a bias and difference in the displacement of magnetically ordered ions in an elementary cell can affect the magnitude of magnetic exchange interactions in different ways, and not necessarily strictly in the direction of their strengthening or weakening. Everything is determined by the features of the overlap and interaction of the electron shells of magnetically active atoms. In a magnetic field with the same induction B = 0.86 Tesla, the values of the specific magnetization and specific susceptibility of the films separated from the substrates of the structures obtained under different conditions were determined at room temperature with different orientations to the external magnetic field. The results of the study averaged over three films are presented in Table 2. Obviously, depending on the conditions for obtaining the structures, the values of the specific magnetic parameters differ markedly. Since all samples were obtained by the same procedure on the same ceramics from the same solution of chemical and electrochemical metallization, other conditions being equal, it is obvious that the change in the specific magnetization of the films in all cases is due to a change in their magnetic structure. The impact of polarization, as well as the impact of an external magnetic field, lead to deformation of the unit cell of the magnetic film. The change in the film structure is due to the most favorable state, from the point of view of entropy conservation during the production process, and is forced to be preserved after the removal of the external fields, both electric and magnetic. Comparison of the magnetization values with the value of the magnetoelectric coefficient for films obtained by different methods show complete agreement with one another. The impact of an external magnetic field during metal deposition leads to a decrease in the values of magnetic characteristics due to deformation of the unit crystal cell. On the contrary, the effect of an external electric voltage during synthesis leads to an increase in the magnetic parameters and cell dimensions. Despite the completely unexplored mechanism of changing the magnetic parameters of the structure, it can be unequivocally stated that the action of external fields (both electric and magnetic) makes it possible to control the magnetic parameters of the films and, to some extent, improve the magnetoelectric parameters.

## 5. Conclusions

The layered Ni-PZT-Ni structures obtained by electrochemical deposition demonstrate good magnetoelectric parameters that are commensurate with the best parameters based on rare earth metals; however, in contrast to them, the magnitude of the displacement in them is much smaller, which allows them to be successfully used to create current sensors and memory elements.

**Author Contributions:** All the authors contributed to this work. Conceptualization, N.P. and V.L.; Data curation, D.F. and N.P.; Formal analysis, A.A. and K.Y.; Funding acquisition, D.F., N.P. and V.L.; Methodology, N.P. and V.L.; Project administration, D.F. and N.P.; Writing—original draft, N.P. and D.F. All authors have read and agreed to the published version of the manuscript.

**Funding:** This research was carried out at the Institute of Technical Acoustics within the framework of basic funding 1.14 «Synthesis, structure and properties of ferroelectric, magnetic and composite materials with high magnetoelectric and dielectric characteristics», Research at Novgorod State University was funded by a grant from the Russian Science Foundation project № 22-19-00763.

**Data Availability Statement:** Not applicable.

**Conflicts of Interest:** There are no conflict of interest. The funders had no role in the design of the study; in the collection, analyses, or interpretation of data; in the writing of the manuscript; or in the decision to publish the results.

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
