# Peer review of "Magnetoelectric Properties of Ni-PZT-Ni Heterostructures Obtained by Electrochemical Deposition of Nickel in an External Magnetic Field"

_magnetochemistry, doi:10.3390/magnetochemistry9040094_

Round 1

Reviewer 1 Report

Referee Report

on paper “Magnetoelectric Properties of NI – PZT – NI heterostructures Obtained by Electrochemical Deposition of Nickel in External Magnetic Field” (magnetochemistry-2289238) by authors Natalia Poddubnaya, Dmitry Filippov, Vladimir Laletin, Andrey Apleksin and Kazimir Yanushkevich, submitted to Magnetochemistry

This is interesting paper. It reports about the correlation between the intensity of the external fields (electric and magnetic fields) and magnetoelectric properties (as function of the orientation) of the metal/ceramic layered structures. It was produced Ni/PZT/Ni samples using electrochemical deposition method. There are several methods and techniques were used for investigations. The obtained experimental results are interesting and reliable. However, paper needs some improvement only after which it can be accepted. At this stage, my decision is minor revision. But I hope that after revision this paper can be accepted in Magnetochemistry. I impressed by the paper.

1.       I recommend revise the title. I propose replace “NI” by the correct symbol “Ni”. But this is not strong require. Only as recommendation (optionally). Authors can use original title.

2.       Abstract is nicely written and clear for understand. But I propose add some significant information what was obtained (some experimental values). I observed that authors demonstrated significant values of the ME coefficients. But this is also recommendation (optional).

3.       Please highlight in Introduction the motivation of the chemical composition of the PZT. What is the composition?

4.       I feel that Introduction can be enhanced. The choice of the research object is attractive. This is heterostructures based on ferroelectric and ferromagnetic materials. I also agree with authors that these structures attract great attention. They are widely used for double-types sensors and memory elements. Please discuss scientific and practical importance of the CerMet structures (doi.org/10.1016/j.ceramint.2019.03.234; https://doi.org/10.1016/j.jmmm.2019.04.006) and the features of the Ni-based electrodeposited materials (10.1016/j.jallcom.2018.03.245; https://doi.org/10.1016/j.ijmecsci.2021.106952) in Introduction.

5.       Please add number for all equation (for example for eq. on page 2 line 73).

6.       Please check the figure captions – revise them for uniform state. For example: on Fig. 2 authors used “mm” and on Fig. 3 “microns”. Also authors used several types of font. There are some typos (please check Fig. 3 – “Figure 3. Figure 3 – Linear…” – why authors used Figure 3 twice?).

7.       On Figure 4 it’s unclear the label of the Y-scale. Please make it correct.

8.       In Tab. 2 (second column “Magnetic parameters”) – I’m not sure that sigma symbol means “magnetic permeability”. As a rule this is “magnetization” And permeability symbol is “m” (not sigma

9.       There are some typos and grammatical errors in the text. Please revise this.

10.   My decision is minor revision. But I impressed by the paper. I feel that after brutal revision it can be accepted.

Author Response

Dear Reviewer,

Thank you for your valuable comments and suggestions that helped improve the quality of our manuscript. According to your comments, we have made the following changes to the text of the manuscript:

Q1)

Abstract is nicely written and clear for understand. But I propose add some significant information what was obtained (some experimental values). I observed that authors demonstrated significant values of the ME coefficients. But this is also recommendation (optional).

We added the numerical value  . The maximum ME voltage coefficient 300 mV/(cmOe) was obtained at transverse orientation at bias magnetic field near 20 Oe.

Q3) . Please highlight in Introduction the motivation of the chemical composition of the PZT. What is the composition?

We added Samples of powdered ceramics PZT composition PbZr0.45Ti0.55O3 are produced …….

 Q4) .       I feel that Introduction can be enhanced. The choice of the research object is attractive. This is heterostructures based on ferroelectric and ferromagnetic materials. I also agree with authors that these structures attract great attention. They are widely used for double-types sensors and memory elements. Please discuss scientific and practical importance of the CerMet structures (doi.org/10.1016/j.ceramint.2019.03.234; https://doi.org/10.1016/j.jmmm.2019.04.006) and the features of the Ni-based electrodeposited materials (10.1016/j.jallcom.2018.03.245https://doi.org/10.1016/j.ijmecsci.2021.106952) in Introduction.

We make some addition and Referencies

Q4) The use of heterostructures based on soft magnetic materials opens their wide use for creating double sensors and memory elements [19-20].

19 Stognij,A.I.; Novitskii, N.N.; Trukhanov, A.V.; Panina S.A.; Sharko,A.I.; Serokurova, A.I.; Poddubnaya N.N.; V.A., Ketsko; Dyakonov V.P.;  Szymczak,H.; Singh g, Charanjeet;  Yang, Y. Interface magnetoelectric effect in elastically linked Co/PZT/Co layered structures Journal of Magnetism and Magnetic Materials, 2019, 485, 291-296

20 Stognij,A.I.; Sharko,A.I.; Serokurova, A.I.; Trukhanov,S.V.; Trukhanov, A.V.; Panina S.A.; Ketsko A.V.; Dyakonov V.P.;  Szymczak,H.; Vinning, D.A.;Singh g, Gudkova,S.A. Preparation and vinvestigation of the magnetoelectrivc properties in layered cermet structures Ceramic International, 2019, 45, 1330-1336

Q5) .       Please add number for all equation (for example for eq. on page 2 line 73).

Now Eg. on page 2 is into text

Q6) We checked caption, the thickness of thin layer usually in μ, and the value of ME coefficient is mV/(cmOe)

Q7) With pity it is impossible. It is the device-scale.

Q8) σ check on μ

Thank you very much for your work

Best regards

Reviewer 2 Report

The authors have performed research and written the corresponding manuscript entitled "Magnetoelectric Properties of NI – PZT – NI heterostructures Obtained by Electrochemical Deposition of Nickel in External Magnetic Field". The topic presented is interesting and can be considered for acceptance after considering the following points:

Abstract: The abstract is clear except that it does not show any result or brief conclusion, usually written in an abstract. Please put the most important results and conclusion at the end part of the abstract.

Figure 4: Please correct the appearance of the y-axis title for both the main graph and the inset.

Figure 5: Could the authors enlarge the font size of the letters inside the Figure to be more readable?

Page 3: Section 3 should only be entitled "Experimental Results" since the discussion has its own section in Section 4.

A conclusion section should be included after the Discussion section.

Could the authors further elaborate on the importance of the research compared to that of the previous papers?

Author Response

Dear Reviewer,

Thank you for your valuable comments and suggestions that helped improve the quality of our manuscript. According to your comments, we have made the following changes to the text of the manuscript:

Q1)

Abstract: The abstract is clear except that it does not show any result or brief conclusion, usually written in an abstract. Please put the most important results and conclusion at the end part of the abstract.

We added the numerical value  . The maximum ME voltage coefficient 300 mV/(cmOe) was obtained at transverse orientation at bias magnetic field near 20 Oe.

Q2) Figure 4: Please correct the appearance of the y-axis title for both the main graph and the inset.

We apologize, but we can not correct, because it is the device-panel

Q3)Figure 5: Could the authors enlarge the font size of the letters inside the Figure to be more readable?

We apologize, we cannot do it

Page 3: Section 3 should only be entitled "Experimental Results" since the discussion has its own section in Section 4.

We added Conclusion

A conclusion section should be included after the Discussion section.

Could the authors further elaborate on the importance of the research compared to that of the previous papers? Yes, for example DOI: https://doi.org/10.1103/PhysRevB.68.132408; J. Appl. Phys. 2005  - V. 97. -  113910; https://doi.org/10.1063/1.2137450; J. of Appl. Phys., 2007, v.102, p.093901 (1-4);

Technical Physics, 2012, Vol. 57, No. 1, pp. 44–47. https://journals.ioffe.ru/articles/viewPDF/26871 ;DOI 10.1007/s00339-014-8430-3

DOI 10.1007/s 00339-015-9443-2 DOI: 10.21883/FTT.2017.05.44371.272

DOI: 10.1134/S106378501703018X     Rev. Sci. Instrum. 90, 015004 (2019);

doi: 10.1063/1.5082833

https://aip.scitation.org/doi/10.1063/1.5082833

https://iopscience.iop.org/article/10.1088/1361-6463/ab01a3/meta

 DOI: 10.1103/PhysRevMaterials.3.044403
